# Absorption-induced transmission in plasma microphotonics

Baheej Bathish[1], Raanan Gad[2], Fan Cheng [2], Kristoffer Karlsson [3], Ramgopal Madugani[3], Mark Douvidzon [4], Síle Nic Chormaic [3] & Tal Carmon[2] ✉

Ionised gas, i.e., plasma, is a medium where electrons-ions dynamics are electrically and magnetically altered. Electric and magnetic fields can modify plasma's optical loss, refraction, and gain. Still, plasma's low pressure and large electrical fields have presented as challenges to introducing it to micro-cavities. Here we demonstrate optical microresonators, with walls thinner than an optical wavelength, that contain plasma inside them. By having an optical mode partially overlapping with plasma, we demonstrate resonantly enhanced light-plasma interactions. In detail, we measure plasma refraction going below one and plasma absorption that turns the resonator transparent. Furthermore, we photograph the plasma's micro-striations, with 35 μm wavelength, indicating magnetic fields interacting with plasma. The synergy between microphotonics and plasma might transform micro-cavities, and electro-optical interconnects by adding additional knobs for electro-optically controlling light using currents, electric-, and magnetic-fields. Plasma might impact microphotonics by enabling new types of microlasers and electro-optical devices.

Integrating plasma in microphotonics might benefit picosecond switches[1] and ultracoherent micro-laser[2] applications, as well as fundamental studies in electron accelerators[3,4], relativistic-[5] and nonlinear-optics[6–8]. In microdevices, plasma was used for illumination, including in display panels[9]. Nevertheless, other than for its radiance, plasma was rarely considered in microphotonics.

Here we fill microbubble cavities[10–14] with plasma to permit resonantly enhanced light–plasma interactions. Using plasma to electro-optically controlling refractive index and absorption in such cavities can enable a new type of microdevice. For example, electrically tuning several resonators to similar resonance frequencies might help commercialise exceptional-point sensors[15,16], sharper top-hat filters, and reconfigurable optical devices. Furthermore, we demonstrate here that absorption results in a seemingly surprising mechanism of absorption-induced transmission that electrically turns on light via a standard fibre. We do that by critically coupling a fibre to a cavity operating near resonance and therefore, even though the cavity

material is highly transparent, light is not transmitted through the fibre. Such devices are generally referred to as coherent perfect absorbers[17–19]. This coherent absorber is destroyed, and the fibre turns transparent, when we invert cavity material from transparent to opaque (here, by electrically igniting plasma).

In a broader view, microresonators with high-quality factors have found numerous applications in ultralow-threshold lasers[20], combs[21,22], gyroscopes[23], detectors[15,16], atomic clocks[24,25], single-photon routers[26], LIDARs[27,28], optical synthesisers[29], cavity quantum electrodynamics[30] and biological microfluidic sensors[31]. Such high-quality factor applications inherently need an ultracoherent light source to serve as their input. Our long-term vision, in this regard, is to enable the mass production of ultracoherent microlasers by combining microresonators with plasma gain. Specifically, plasma at population inversion[2] (not demonstrated here) in the evanescent region of an ultrahigh-Q cavity has the potential to transform such microresonators into electrically pumped lasers. Such lasers might serve as the necessary ultracoherent

[1]Faculty of Mechanical Engineering, Technion-Israel Institute of Technology, Haifa 3200003, Israel. [2]School of Electrical Engineering, Tel Aviv University, Tel Aviv 6997801, Israel. [3]Okinawa Institute of Science and Technology Graduate University, Onna, Okinawa 904-0495, Japan. [4]Solid State Institute, Technion-Israel Institute of Technology, Haifa 3200003, Israel. ✉e-mail: total@tauex.tau.ac.il

source in experiments such as those described in refs. [15,16,20–31] and transform them into daily-life applications.

Here we introduce plasma to microcavities. Then, using our plasma-containing microcavity, we show absorption-induced transparency of these high-Q fibre-coupled microphotonics. In addition, we show that plasma electrically changes absorption and refraction of high-Q microphotonics.

The schematic description of our electro-optical system includes plasma in the inner part of a microbubble resonator[10–14] (Fig. 1a). The microbubble cavity resonantly enhances light when its circumference is an integer number of optical wavelengths. Crucially, a significant part of the resonance's mode-volume overlaps with the inner volume of the microbubble, where plasma resides, allowing us to electrically change the resonance properties by electrically ionising argon gas, creating plasma or by plasma recombination processes.

In all the microresonators that have been used by now, the light propagates in solid, liquid or gas media. Here, we demonstrate a plasma-filled micro-resonator in which the optical resonance partially overlaps with plasma. The design and fabrication of our microbubble resonator optimises light extension into the plasma region. The

resonator wall thickness is about $1\,\mu m$ through pretapering a silica capillary, then heating using a $CO_2$ laser while controlling applied inner air pressure (see "Microbubble cavity fabrication"). To further evanescently couple light into the inner plasma, we use a relatively long optical wavelength at $1.55\,\mu m$. For similar reasons, we use a relatively large resonator radius of $90\,\mu m$, thus reducing the tendency of light to centrifugally move away from the inner plasma. We then fill the microbubble cavity with argon gas and insert micro copper electrodes through both sides of the capillary. Sharp-tipped electrodes are used to enhance the local electric field to achieve gas breakdown.

In detail, assuming Lorentzian absorption that characterises resonators, the calculated optical transmission through the fibre is[32]

$$T = 1 - \frac{4(\delta_c)(\delta_0 + \widetilde{\delta_p})}{\left(\delta_c + \delta_0 + \widetilde{\delta_p}\right)^2 + \left(\Delta\omega + \widetilde{\Delta\omega_p}\right)^2} \quad (1)$$

where $\delta_0$ and $\delta_c$ represent the cavity coupling and loss rates, and $\Delta\omega$ is the angular-frequency detuning between the laser and the cold cavity resonance. Plasma results in an additional cavity loss rate, $\widetilde{\delta_p}$, and

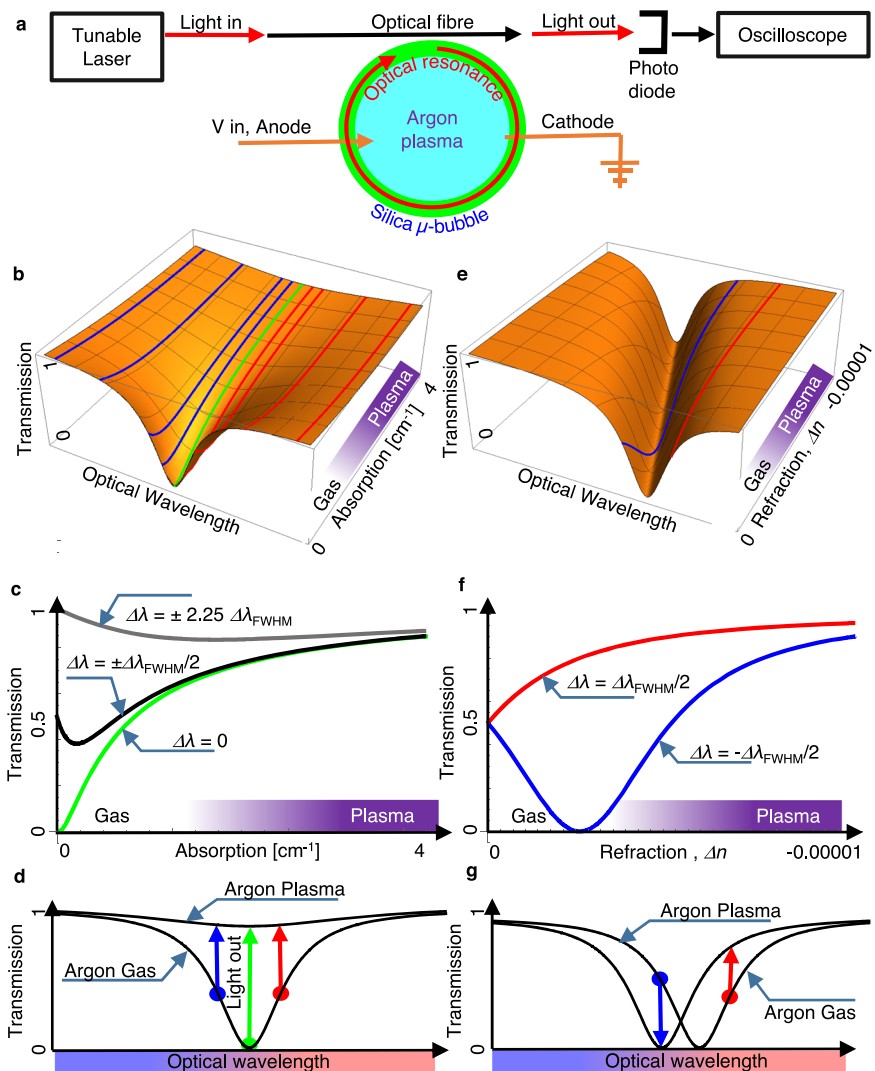

**Fig. 1 | Schematic description of the experimental system. a** The plasma-containing microcavity. **b–g** Calculated transmission through the fibre-coupled plasma resonator as a function of changes in the plasma's absorption (**b–d**) and refractive index (**e–g**). We assume Lorentzian resonance absorption. Absorption-induced transmission is evident in (**b–d**), as indicated by the green lines and arrow in (**b, c**). Absorption-induced effects are symmetrical with detuning, as indicated by the blue and red lines and arrows in (**b, d**). On the contrary, index-induced effects for blue or red detuning are different, as indicated by the blue and red lines and arrows in (**e–g**).

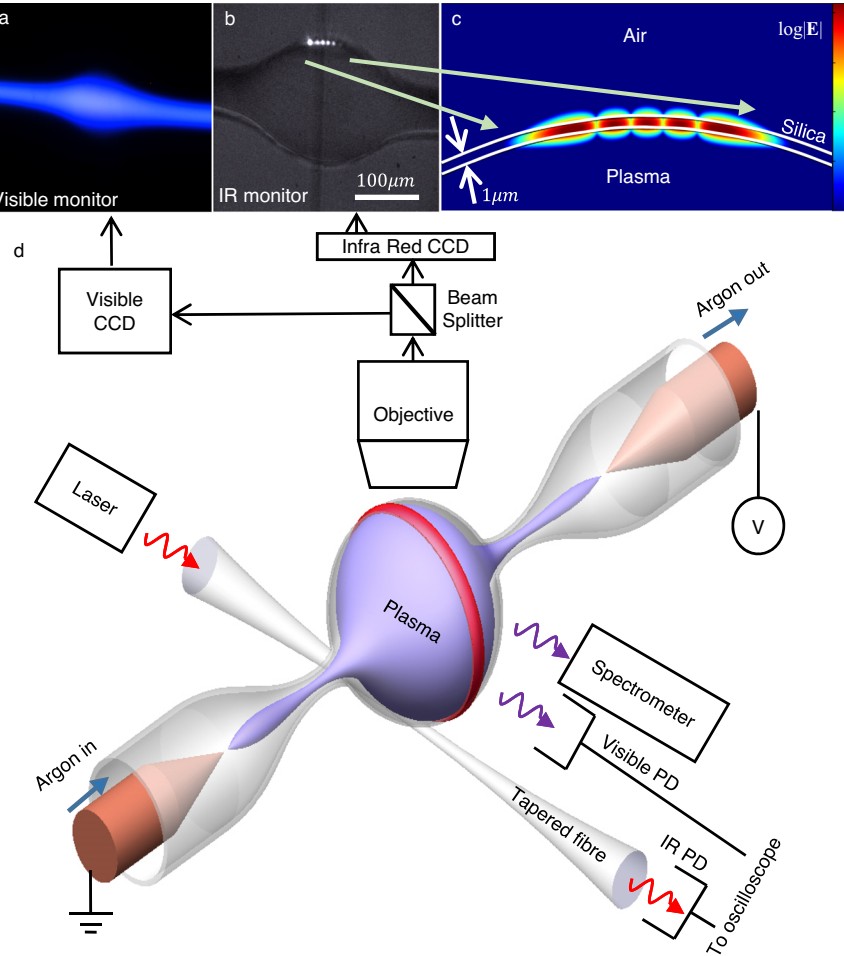

**Fig. 2 | Experimental setup. a** Micrograph of the plasma's glow taken using a visible camera. **b** Micrograph of the optical resonance taken using a camera sensitive to 1.55 μm wavelength. The five spots at the top part of the bubble indicate the transverse cross section of the 5th order transverse mode, as indicated by residual forward scattering. **c** Numerical calculation of the 5th order polar mode reveals 4% penetration of the resonating power into the plasma-containing region. **d** Drawing of the experimental system. A silica capillary tube is filled with argon. Connected with a high voltage source, two electrodes (two orange cones) are inserted into the silica capillary tube to active plasma. Laser light can be coupled into the bubble resonator through a tapered fibre. The red circle represents the optical mode. On the other side of the fibre, an infrared photodiode (PD) is used to monitor the transmission. A visible photodiode is used to monitor plasma luminescence. A spectrometer is used for plasma emission spectroscopy. A visible camera is used for inspecting plasma luminescence and an infrared camera is used for inspecting the infrared resonance.

detuning, $\widetilde{\Delta\omega_p}$, where the ~ stands for their time variation in our experiment. The additional cavity loss rate, $\widetilde{\delta_p}$, and detuning, $\widetilde{\Delta\omega_p}$, are a function of plasma loss, $\alpha_p$, and refractive index, $\Delta n_p$, which depends on plasma properties (see "Theory of resonantly enhanced light–plasma interaction"). As evident from Eq. (1), increasing plasma absorption, $\delta_p$, increases the transmission through the cavity. Near critical coupling ($\delta_c = \delta_0$), transmission can rise from 0 to almost 1 as plasma absorption rises (Fig. 2b, green). While critically coupled, increasing the absorption of the cavity material takes the cavity to the under-coupled regime, thereby reducing cavity absorption and consequently switches on the light emitted through the fibre. This absorption-induced transmission occurs since the higher optical loss is associated with a lower cavity quality factor that takes the fibre-coupled cavity to its under-coupled[32] regime. While under-coupled, the effect of the resonator on the nearby fibre waveguide is negligible, so the fibre turns almost transparent.

As for changes in refractive index, they cause a drift in the resonance frequency to increase (or reduce) transmission, depending on whether initial detuning conditions are to the red or the blue sides of the resonance wavelength (Fig. 2e–g, red and blue lines and arrows).

Here, we evanescently couple continuous wave [CW] laser light to the bubble resonator through a tapered fibre coupler[33] and monitor transmission by connecting the other side of the fibre coupler to a photodiode (Fig. 2, New Focus, model 1811). Another photodiode (New Focus, model 1801) sensitive to visible light is coupled through free space to monitor plasma luminescence (Fig. 2). A spectrometer sensitive to visible and near-infrared [IR] is simultaneously used for plasma emission spectroscopy. In addition, we use a visible camera for inspecting plasma luminescence (Fig. 2a) and simultaneously, an IR camera inspecting the IR resonance (Fig. 2b) via its residual forward scattering. High-order TM bubble modes are preferred here (see "Mode mapping") since they better penetrate the plasma[31]. For example, the 5th order polar mode (Fig. 2b) has 4% of its mode-volume overlapping with plasma, as calculated in "Theory of resonantly enhanced light–plasma interaction" and shown in Fig. 2c. As for the coupling, the fibre touches the resonator to improve stability. Two electrodes (electrical wires) are inserted into the silica capillary tube while the gap between the electrodes and the glass is sealed using glue. Our following experiments measure optical transmission through the cavity at different detuning while the plasma is ignited.

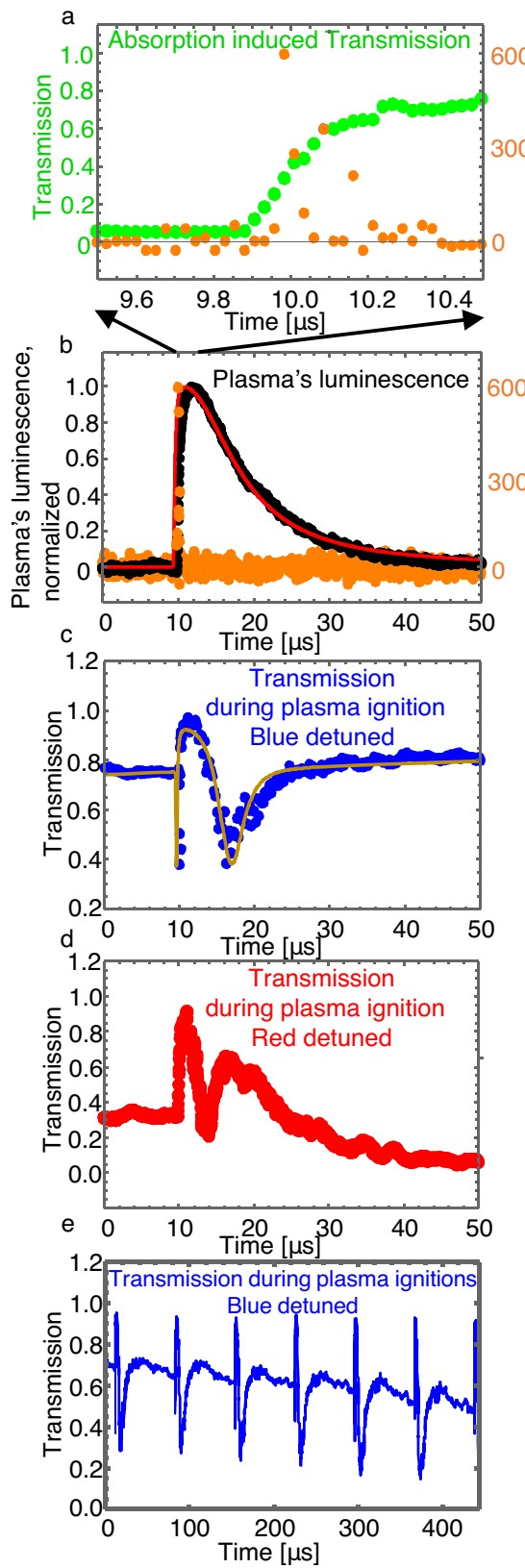

**Fig. 3 | Experimental result. a** Absorption-induced transmission: a voltage spike (orange) indicates the electrical breakdown of argon gas to plasma. This breakdown is accompanied by an increase in transmission (green) resulting from plasma's absorption. **b** Plasma luminescence (black) which we use to estimate the plasma's density. Red: assumed exponential rise and decay. **c** Blue-detuned transmission during plasma ignition shows a dip followed by a peak change governed by absorption followed by a refraction-induced drift that reduces transmission. Blue: experimental results. Line: best fit using Eq. (1). **d** Red detuned transmission during plasma ignition shows, as expected, absorption-induced effects identical to (**c**) and refraction-induced effects opposite to (**c**). **e** Repeatability and controlled detuning. We can repeat plasma ignition as many times as needed with repeatability as appears in the plot. Note that the slight overall downward slope in transmission is indicating an intentional slow continuous detuning of the laser frequency in respect with the thermally broadened resonance frequency of the cold cavity. In this manner, we can scan to any desired region at the resonance sidebands. **b**, **c** were measured simultaneously. The circle size represents the estimated error.

power is 100 ns while plasma breakdown is characterised by a voltage rise time of ~20 ns (Fig. 3a, orange) for 430 volts applied via electrodes separated by 5 mm. The relatively long response time here, when compared to electro-optical switches[34,35], is the result of the large separation between the electrodes (5 mm). Reducing the electrode distance to 50 microns may facilitate similar optical switching times using a standard 5 V source. Previously, plasma-based electrical switches were reported to respond within 10 ps[1] when the distance between electrodes was as short as 3 microns. Moreover, unlike MEMS and thermally based switches where mass inertia and thermal time constants affect their performance[34,35], plasma ignition speed is a function of the electrical power-source characteristics, the distance between electrodes, and pressure, all of which could be optimised for a faster response.

We repeat the transmission measurements, but now while the laser is initially detuned to the red or blue sidebands of the resonance, and not to the resonance centre. We expect that absorption will change transmission regardless of whether detuning is to the blue or red sides of the resonance (Fig. 1d, red and blue arrows). On the contrary, we expect that the changes in the refractive index will result in opposite changes in transmission, depending on the detuning direction (Fig. 1g, blue and red arrows). When the laser wavelength is at the blue side of the resonance, plasma formation (Fig. 3b) results in a narrow transmission dip followed by a peak and then another dip (Fig. 3c). As expected from Fig. 1b–g, the dip can be the result of absorption, while the peak that follows can be explained by changes in the refractive index. This qualitative explanation is followed here by fitting our measured transmission (Fig. 3c, dots) to that of Eq. (1) (Fig. 3c, line) while taking the plasma's absorption and refraction as free parameters. We achieve the best fit to our experimentally measured transmission with plasma absorption $\alpha_p = 3.68\,cm^{-1}$ and plasma refractive index of $0.999993 \pm 0.000001$ for a cavity $Q$ of $2 \times 10^7$. We estimate the error in the refractive index via the largest n deviation that still maintain our fit (Fig. 3c, line) within the range of the experimental error (that is represented by the size of the points in Fig. 3c).

This refractive index corresponds to a plasma frequency of $\omega_p = 4.55 \times 10^{12}$ Rad/s and free-electron density of $n_e \sim 6 \times 10^{15}\,cm^{-3}$ (see "Theory of resonantly enhanced light–plasma interaction"). In our fitting model, plasma absorption and refraction are assumed to change with plasma density, and plasma density is measured via its luminescence using a photodiode. The plasma density is assumed here to follow the plasma's luminescence but to fall faster due to a faster plasma recombination rate near the solid cavity walls[36].

In order to verify the plasma's electron density, we simultaneously measure the optical spectrum of the plasma (see "Plasma emission spectrum") as emitted to a spectrometer through free space (Fig. 2d). As expected from the independent absorption-based

## Results

We start by measuring absorption-induced transmission by tuning our laser wavelength to the cavity's optical-resonance wavelength. The plasma pressure is 2.5 Torr. As expected from theory (Fig. 2b–d, green), transmission rises from 5 to 78% (Fig. 3a, green) upon breakdown and plasma formation. The optical rise time to half-max

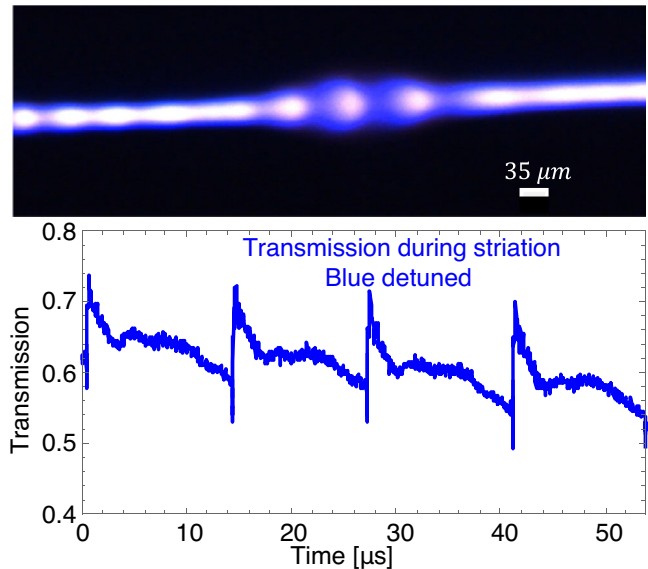

**Fig. 4 | Plasma standing striation.** Micrograph of plasma ionisation wave. Bottom: Transmission during four plasma ignition events while laser is at the blue-detuned side of the resonance. A slight overall slope indicates an intentional continuous detuning of laser frequency.

measurement that appears in the previous paragraph, the relative intensity of Plasma's 2p6-1s5 and 2p2-1s3 lines gives plasma density $n_e \leq 1 \times 10^{16}$ cm$^{-3}$ (see "Plasma emission spectrum").

We will now switch to the red detuned side of the resonance where the absorption-dominant region is expected to remain unchanged (Fig. 1d). In contrast, in the refraction-dominant region the transmission structure is expected to invert (Fig. 1g), as we indeed observe (Fig. 3d). We could repeat plasma ignition events as many time as needed with repeatability as one can see in Fig. 3e.

In the next experiment (Fig. 4), we increase the pressure to 5 Torr and observe bright and dark plasma fringes, which represent regions of higher or lower plasma density. This phenomenon is generally referred to as striations[37]. Striations are the result of electron kinetics within plasmas that conduct electric current. For example, magnetic fields induced by currents affect electron dynamics and play a role in the formation of striations. The smallest striations measured in our system have a 35-μm wavelength.

## Discussion

In conclusion, we experimentally demonstrate plasma-filled microcavities. This new type of microcavity paves the way to plasma microphotonics, where one can use electric and magnetic properties to control optical refraction, absorption and gain. Our results have a broader impact by heralding a new type of plasma-based electro-optical interconnects and high coherency electrically pumped microlasers that might be compatible with both on-chip electronics and on-chip vacuum cells[38]. Furthermore, harnessing the ultrahigh Q of cavities to microplasma might improve the resolution of magnetic field sensors.

## Methods

### Microbubble cavity fabrication

The microbubble cavities were fabricated[39] from commercial fused silica glass microcapillaries (Polymicro Technologies TSP250350) with initial inner and outer diameters of 250 μm and 350 μm, respectively. As shown in Supplementary Fig. 1a microcapillary was clamped at one end to a stationary mount and the other end to a motorised 1D stage (Thorlabs model, MTS50/M-Z8). A high-power $CO_2$ laser (Synrad series model, 48-2KWM with 25 W maximum power), was split to make two

counter-propagating beams and focussed onto a section of the glass capillary. A moderate power of around 10% (2.5 W) focused onto the capillary generates enough heat to burn off the protective outer metal coating, leaving bare capillary. At 20% of $CO_2$ laser power, the glass softened reaching around 1700 °C. The power was slowly increased (to compensate for the power to reach the reducing device surface area), while the motorised stage pulled the capillary, tapering it until the outer diameter reached 30–40 μm range. The $CO_2$ laser was turned off as the stage came to rest. Pressurised gas ($N_2$ gas at 3 Bar) was introduced into the capillary, and the laser was turned on at around 30% power and was very slowly increased until (around 35%) the walls of the capillary became hot and soft enough to allow the pressurised gas to expand it, forming a bubble. While carefully monitoring the laser power, the bubble was allowed to expand further, until the desired diameter, typically in the range of 80–200 μm. The exact dimensions for each bubble were determined after fabrication, using a microscope. Bubble wall thickness can be estimated using the equation[40], $w = a - \sqrt{a^2 - (d^2(1-f^2)/4)}$, where a is the outer diameter of the bubble, d is the tapered capillary outer diameter, f=0.7 is the constant ratio between the inner and outer diameters (*ID/OD*) of the capillary. The optical Q-factors of the bubbles typically range between $10^5$ and $10^7$ at 1550 nm.

### Theory of resonantly enhanced light–plasma interaction

In what follows, we assume that the plasma affects the light through optical refraction and absorption and that light does not affect the plasma. In addition, we analyse the properties of the plasma in the region that overlaps with the optical mode. It is possible that the plasma density in this region is lower than that of other regions inside the microbubble due to double sheet formation and potential drop induced by the silica wall and charge separation[41].

Using the steady-state approximation, the optical transmission through an optical resonator is

$$T = 1 - \frac{4(\delta_c)(\delta_0 + \widetilde{\delta_p})}{\left(\delta_c + \delta_0 + \widetilde{\delta_p}\right)^2 + \left(\Delta\omega + \widetilde{\Delta\omega_p}\right)^2} \qquad (2)$$

where $\delta_0$ and $\delta_c$ are cavity coupling and loss rates. $\Delta\omega$ is the angular-frequency detuning between the laser and the cold cavity resonance frequency. Plasma results in an additional loss rate, $\widetilde{\delta_p}$, and frequency detuning, $\widetilde{\Delta\omega_p}$, where ~ stands for time variation.

The time-varying loss and detuning that plasma induces are

$$\widetilde{\delta_p} = \delta_p D_p(t)^l \text{ and } \widetilde{\Delta\omega_p} = \Delta\omega_p D_p(t)^l \qquad (3)$$

where $D_p(t)$ is plasma density as deduced from plasma luminescence. The plasma density is normalised to have a maximal value of 1 at maximum luminescence. l is a free parameter describing a faster decay of plasma near the bubble wall because of faster recombination of the plasma to a gaseous state of matter[42]. l=3.3 gave the best fitting for our experimental results (Fig. 3c).

Plasma's loss rate and frequency detuning at their maximal value are

$$\delta_p = \frac{\mu_p \alpha_p c}{2n}, \quad \Delta\omega_p = \frac{2\pi c \mu_p \Delta n_p}{\lambda_0 n} \qquad (4)$$

where $\mu_p = 0.04$ is the calculated[43] fraction of optical power that propagates in the region where plasma resides, c and $\lambda_0$ are the vacuum speed and wavelength of light, $\alpha_p$ is plasma absorption, $\Delta n_p$ is plasma refractive index and n is the effective refractive index of the resonance.

The change in refraction and absorption is[44]

$$\alpha_p = \frac{\nu_e \omega_p^2}{2c\omega_0^2}, \Delta n_p = \sqrt{1 - \frac{\omega_p^2}{\omega_0^2}} - 1 \cong -\frac{\omega_p^2}{2\omega_0^2} \tag{5}$$

where $\nu_e$ is plasma recombination rate and $\omega_0$ is the angular optical frequency and $\omega_p$ is plasma frequency.

$$\omega_p^2 = \frac{e^2 n_e}{\varepsilon_0 m_e} \tag{6}$$

where $e$ and $m_e$ are the charge and mass of the electron, $n_e$ is free-electron density, and $\varepsilon_0$ is vacuum permeability.

## Mode mapping

We prefer high-order modes since they evanescently couple better to the inner plasma. Related to the relatively low inner refractive index, we did not see any high-order radial modes, as they frequently appear in water-filled bubbles. The order of our modes was measured to change between 1 and 5 along the polar angle (see Supplementary Fig. 2). We characterise our bubble resonator using a mode mapping technique based on residual Rayleigh scattering to find these high-order modes. For mapping the optical modes, we sweep our laser frequency across bubble resonances. At the same time, we photograph the transverse cross section of the mode using forward scattered light that is imaged by a near-infrared camera (Edmund Optics, 1460–1600 nm Near-Infrared Camera). According to our experience, every optical mode has such forward scattering that is sufficient for imaging the mode structure, irrespective of the mode quality factor. Deviations from a perfectly uniform scatterer distribution might explain slight differences between the micrographs and their corresponding calculated modes.

## Plasma emission spectrum

To confirm plasma electron density as measured via optical refraction and absorption, we conduct a simultaneous independent measurement of electron density. The process we follow here involves concluding on plasma's electron density from its spectral lines, as appears in the following references[45–47]. In this regard, we measure argon spectral lines $\lambda_1 = 763.5$ [nm]:2p6-1s5 and $\lambda_2 = 772.4$ [nm]:2p2-1s3 (see Supplementary Fig. 3) and calculate their relative power. We assume corona steady-state equilibrium; that is, ion excitation occurs due to electron collisions. Also, we expect de-excitation by spontaneous emission, which means that the plasma is optically thin, and photons mostly escape without interacting with neutrals and ions. For an electron temperature Te - 1[eV], and using the procedures explained in refs. 45–47, we get plasma electron density $n_e \leq 1 \times 10^{16}$ cm$^{-3}$, which agrees with what we got from plasma absorption and refraction (see "Microbubble cavity fabrication").

## Data availability

The data presented in this study have been deposited in a Zenodo repository at https://doi.org/10.5281/zenodo.8024815.

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

## Acknowledgements
This research was supported by the United States–Israel Binational Science Foundation (NSF-BSF), Grant No. 2020683 (T.C.), and the Israeli Science Foundation, Grant No. 537/20 (T.C.) and Okinawa Institute of Science and Technology Graduate University.

## Author contributions
B.B. performed the experiments. F.C. performed the theoretical analysis. K.K. and R.M. fabricated microbubble cavities. R.G., M.D. and S.N.C. supervised the work. T.C. supervised all aspects of the work.

## Competing interests
The authors declare no competing interests.
