## [Peer Review File · Nature Communications]

Absorption-Induced Transmission in Plasma MicrophotonicsREVIEWER COMMENTS

Reviewer #1 (Remarks to the Author):

In their manuscript the authors investigate a hollow microresonator filled with an Argon plasma. To my knowledge this is the first ever investigation of the evanescent field of a microresonator with a plasma and makes a big contribution for further investigation of this exciting line of research. The methodology is sound, and the results are well presented and with sufficient detail to replicate. Maybe a line or two on how they sealed the electrodes in to the bubble resonator might be useful..

I find the work of high quality and suitable for Nature Communication, as it realised for the first time a compact way to study the effects of evanescent resonant light interaction in a microresonator with a plasma opening a new and exciting field of research.

Minor Points:

- My only concern is the motivation of using the plasma as an electronic switch, which it clearly is, but I would like the authors to clearly identify the differences between such switching versus switching via either piezo/pressure tuning of the resonance of the microresonator, or nonlinear-electro optic tuning, or temperature tuning, all these methods change the refractive index and thus can lead to a switching and it seems to me that the timescale for igniting the plasma might be longer than some of the other methods.
- I also wonder about the significant digits given for the refractive index that is measured below 1, how certain are the authors of these values?

Reviewer #2 (Remarks to the Author):

The manuscript presents original results from an experiment dedicated to the modification of the optical properties of a hollow microcavity due to plasma ignition of a rare gas within it. More precisely, the authors show that the resonance properties of the microcavity are modified by the plasma, such that the evanescent coupling to an external optical fiber can be modulated, and thus its transmission.

Microbubbles created in a capillary tube by laser heating have been widely used for more than 10 years for sensing molecules or other physical parameters (magnetic field, pressure, flow rates, etc) in microfluidics devices (since Sumetsky et al Opt. Lett. 2010, 35, 898–900), using similar shifts of the high Q WGM modes of the microbubbles. Generating plasma in microcavities has also been extensively

studied and carried out, even in large panels of thousands of microcavities for illumination purposes. The current manuscript combines these 2 well developed technologies to demonstrate electrically triggered transmission switching in an optical fiber. Notice that an interesting paper using plasma microbubbles in capillary tubes for optimizing phase matching of high harmonics generation, making use of the same refraction index switch due to the presence of plasma, was published as early as

The manuscript, however, focuses more on possible use of the technique and speculative applications than on the scientific processes involved in the experiment. For example, in the introduction, half a page describes the possibility of using this technology for producing microlasers in the future, which appears unrelated to the presented experiment. The manuscript also suffers from some “overselling” statements or unnecessary hype, e.g. “here we introduce plasma, known as the fourth state of matter, to microcavities” (!!), “increasing the absorption of the cavity material reduces cavity absorption” (!!), “we will now perform a control group experiment by switching to the red side of the resonance” (!!), “we experimentally demonstrate plasma filled microcavities for the first time “ (!!), etc..

I also doubt that a gas filled HV-switched device with 100 ns rise time will be heavily used on photonic platforms, in contrast to what is repeatedly claimed in the manuscript.

Most of the scientific argumentation is also lacking. Basic simulations on WGM resonances shifted by a complex refractive index modification are presented, without the specificity of light to plasma coupling, surface-plasma interaction, etc. Also the phenomenon of “striation” is described as a highlight when the pressure is increased, but no convincing explanation of the phenomenon is presented.

In summary, the manuscript can't be published in Nature Communications in the present form. It would require a substantial (achievable ?) improvement of the scientific case, less generalities and more realistic statements to be re-considered.

Reply to referees

We are glad that Referee 1 finds our work “*of high quality and suitable for Nature Communication, as it realised for the first time a compact way to study the effects of evanescent resonant light interaction in a microresonator with a plasma opening a new and exciting field of research.*”

While Referee 2 agrees that our “*manuscript presents original results...*”, they ask for “*substantial improvements*”.

We have revised the manuscript according to the suggestions and critics of the Reviewers. In the following, we answer and discuss the comments of Reviewers #1 and #2. For the sake of clarity, we will first include the Reviewer's comment and then respond. A marked version is also attached. Added text is marked in green and deleted text is marked in red.

We hope that the present version of the manuscript can be accepted.

Reply to Referee 1:

We thank the referee for their professional report.

In what follows, we addressed all the points raised by Referee 1.

Q1: The Referee asks for details on sealing the electrodes that were going into the bubble resonator.

A1: We accordingly added:

“The electrical wires were inserted into the silica capillary tube while the gap between the wire and the glass was sealed using glue.”

Q2: I would like the authors to clearly identify the differences between such switching versus switching via either piezo/pressure tuning of the resonance of the microresonator, or nonlinear-electro optic tuning, or temperature tuning, all these methods change the refractive index and thus can lead to a switching. The timescale for igniting the plasma might be longer than some of the other methods.

A2: We accordingly revised our text to:

“The relatively long response time here, when compared to electro-optical switches^{38,39}, is the result of the large separation between the electrodes (5 mm). Reducing the electrode distance to 50 micron may facilitate similar optical switching times using a standard 5 Volts source. Previously, plasma-based electrical switches were reported to respond within 10 picosecond¹ when

the distance between electrodes was as short as 3 microns. Moreover, unlike MEMS and thermally based switches where mass inertia and thermal time constants affect their performance^{38,39}, plasma ignition speed is a function of the electrical power-source characteristics, the distance between electrodes, and pressure, all of which could be optimized for a faster response.”

We have added references 38, and 39 that compare the performance of many optical switching methods, among them thermal, MEMS, electro-optical, and acousto-optical ones. As appeared in references 38, and 39, it takes thermal and MEMS switches typically longer than one ms to respond.

Reference 38: "Ma, X. & Kuo, G.-S. Optical switching technology comparison: optical MEMS vs. other technologies. *IEEE Commun. Mag.* **41**, S16–S23 (2003). "

Reference 39: "El-Bawab, T. S. *Optical Switching*. (Springer Science & Business Media, 2008)."

Q3: I also wonder about the significant digits given for the refractive index that is measured below 1, how certain are the authors of these values?

A3: We have revised our text to:

“...plasma refractive index of 0.999993 ± 0.000001 for a cavity Q of 2×10^7 . We estimate the error in the refractive index via the largest n deviation that still maintain our fit (Fig 3c, line) within the range of the experimental error (that is represented by the size of the points in Fig 3c).”

As appeared in our manuscript, we measured this index reduction at the red and blue detuned sides of the resonance. We note that thermal expansion and thermal coefficient of refractive index are such that optical path length increases with temperature. In this regard, the only effect that could reduce refractive index, as we measured, is plasma – as plasma theory predicts.

Reply to Referee 2:

We thank the referee for their comprehensive report.

Q1:” Notice that an interesting paper using plasma microbubbles in capillary tubes for optimizing phase matching of high harmonics generation, making use of the same refraction index switch due to the presence of plasma, was published as early as...”

A1: Comment is well noted, but unfortunately the final sentence seems to have been cut so we are not fully sure of the reference made. We have added the reference: “Tenio Popmintchev, Brendan Reagan, David M. Gaudiosi, Michael Grisham, Mark Berrill, Oren Cohen, Barry C. Walker, Margaret M. Murnane, Henry C. Kapteyn, and Jorge J. Rocca. High-order harmonic generation from ions in a capillary discharge. *Phys. Rev. Lett.* **96**, 203001 (2006).”

Q2: “in the introduction, half a page describes the possibility of using this technology for producing microlasers in the future, which appears unrelated to the presented experiment”

A2: The referee has a point. We have shortened the half page description to:
“Our long-term vision is to enable the mass-production of ultracoherent microlasers by combining microresonators with plasma gain. Specifically, plasma at population inversion² (not demonstrated here) in the evanescent region of an ultrahigh-Q cavity has the potential to transform such microresonators into electrically pumped lasers. Ultracoherent lasers, such as those described in¹⁷⁻³⁰, are crucial light sources for real-world applications¹⁷⁻³⁰, where the ultrahigh-Q resonator serves as the device's core component.”

Q3: Referee suggests not to use overselling.

A3.1: We revised:

“Here we introduce plasma, known as the fourth state of matter, to microcavities” to
“Here we introduce plasma to microcavities”

A3.2: We revised:

"increasing the absorption of the cavity material reduces cavity absorption" to

“While critically coupled, *increasing* the absorption of the cavity material takes the cavity to the under coupled regime, thereby *reducing* cavity absorption”

A3.3: We revised:

"we will now perform a control group experiment by switching to the red side of the resonance" to:

“We will now switch to the red detuned side of the resonance”

A3.4: We revised:

"we experimentally demonstrate plasma filled microcavities for the first time " to

"we experimentally demonstrate plasma filled microcavities "

We removed "as known to us, these are among the smallest striations ever observed."

Q4: I also doubt that a gas filled HV-switched device with 100 ns rise time will be heavily used

A4: After consideration of this comment, we have revised our text to:

"The optical rise-time to half-max power is 100 ns while plasma breakdown is characterised by a voltage rise time of ~20 ns [Fig. 3a Orange] for 430 volts applied via electrodes separated by 5 mm. The relatively long response time here, when compared to electro-optical switches^{34,35}, is the result of the large separation between the electrodes (5 mm). Reducing the electrode distance to 50 micron may facilitate similar optical switching times using a standard 5 Volts source."

Q5: Basic simulations on WGM resonances shifted by a complex refractive index modification are presented, without the specificity of light to plasma coupling, surface-plasma interaction, etc.

A5: The focus was on the observations. However, we have added:

"In what follows, we assume that the plasma affects the light through optical refraction and absorption and that light does not affect the plasma. Additionally, we analyse the properties of the plasma in the region that overlaps with the optical mode. It is possible that the plasma density in this region is lower than that of other regions inside the micro bubble due to double sheet formation and potential drop induced by the silica wall and charge separation⁴⁴."

Q6: "striation" is described as a highlight when the pressure is increased, but no convincing explanation of the phenomenon is presented.

A6: We have added some description to make this clearer to the reader. We have revised the text:

"In the next experiment [Fig. 4], we increase the pressure to 5 Torr and observe bright and dark plasma fringes, which represent regions of higher or lower plasma density. This phenomenon is generally referred to as striations⁴¹. Striations are the result of electron kinetics within plasmas that conduct electric current. For example, magnetic fields induced by currents affect electron dynamics and play a role in the formation of striations."

Q7: Generating plasma in microcavities has also been extensively studied and carried out, even in large panels of thousands of microcavities for illumination purposes.

A7: We accordingly added:

"Plasma was used for illumination, including in display panels³¹."

REVIEWERS' COMMENTS

Reviewer #1 (Remarks to the Author):

The authors did a good revision of their manuscript and have answered all of my questions satisfactorily. I therefor can recommend the publication of this manuscript in NC.

Reviewer #2 (Remarks to the Author):

I thank the authors for having taken into account all my remarks and answering all my questions in a very satisfactory way. From my side, the MS can now be published in Nature Communications.
Congratulations

Reply to referees

Referee 1 writes that “The authors did a good revision of their manuscript and have answered all of my questions satisfactorily. I therefor can recommend the publication of this manuscript in NC.”

Referee 2 writes that “thank the authors for having taken into account all my remarks and answering all my questions in a very satisfactory way. From my side, the MS can now be published in Nature Communications. Congratulations”

Referee 1 and 2 have no points to address.

We thanks the referees for their efficient review.